# Effect of Mitotane on Male Gonadal Function

**DOI:** 10.3390/cancers15123234

**Published:** 2023-06-18

**Authors:** Federica Innocenti, Sara Di Persio, Marilena Taggi, Roberta Maggio, Pina Lardo, Vincenzo Toscano, Rita Canipari, Elena Vicini, Antonio Stigliano

**Affiliations:** 1Department of Anatomy, Histology, Forensic Medicine and Orthopedic, Section of Histology, Sapienza University of Rome, 00185 Rome, Italy; innocenti@generalifeitalia.it (F.I.); sara.dipersio@ukmuenster.de (S.D.P.); marilenataggi@gmail.com (M.T.); rita.canipari@uniroma1.it (R.C.); 2Endocrinology, Department of Clinical and Molecular Medicine, Sant’Andrea University Hospital, Sapienza University of Rome, 00185 Rome, Italy; r.maggio@uniroma1.it (R.M.); pina85la@gmail.com (P.L.); vincenzo.toscano@uniroma1.it (V.T.)

**Keywords:** mitotane, adrenocortical carcinoma, testis, testosterone, male gonadal function

## Abstract

**Simple Summary:**

Mitotane (MTT) is the treatment of choice for adrenocortical carcinoma. Male hypogonadism is often diagnosed in male patients treated with this drug. This research aims to consider possible side effects induced by MTT in the testis and a hypothetical detrimental effect deriving from androgenic deficiency. Furthermore, considering the increased potentiality of treatment for adrenocortical carcinoma, we want to provide male patients of childbearing age, as in other oncological diseases, the opportunity to consider a sperm cryopreservation program.

**Abstract:**

Background: Clinical evidence has shown frequent hypogonadism following mitotane (MTT) treatment in male patients with adrenocortical carcinoma. This study aimed to evaluate the impact of MTT on male gonadal function. Methods: Morphological analysis of testes and testosterone assays were performed on adult CD1 MTT-treated and untreated mice. The expression of key genes involved in interstitial and tubular compartments was studied by real-time PCR. Moreover, quantitative and qualitative analysis of spermatozoa was performed. Results: Several degrees of damage to the testes and a significant testosterone reduction in MTT-treated mice were observed. A significant decline in *3βHsd1* and *Insl3* mRNA expression in the interstitial compartment confirmed an impairment of androgen production. *Fsh-R* mRNA expression was unaffected by MTT, proving that Sertoli cells are not the drug’s primary target. Sperm concentrations were significantly lower in MTT-treated animals. Moreover, the drug caused a significant increase in the percentage of spermatozoa with abnormal chromatin structures. Conclusion: MTT negatively affects the male reproductive system, including changes in the morphology of testicular tissue and reductions in sperm concentration and quality.

## 1. Introduction

Adrenocortical carcinoma (ACC) is a rare and aggressive endocrine tumor with poor oncological and endocrinological prognoses [1]. ACC affects about two million people per year with a bimodal incidence and a peak incidence in pediatric and middle ages [1]. The clinical picture is frequently characterized by high tumor cortisol secretion and sometimes androgens with the typical signs of overt Cushing’s syndrome and a more complicated endocrinological status [1]. In fact, the excess of cortisol interferes with the physiological effects of other hormones and endocrine pathways [2]. This condition adversely impacts many metabolic processes involving the hypothalamic-pituitary-gonadal axis, resulting in hypogonadotropic hypogonadism [2].

Mitotane (1,1-dichloro)-2-(*o_*-chlorophenyl)-2-(*p_*-chlorophenyl)-ethane or *o_′,p_′*-DDD) (MTT) is an adrenolytic drug widely used in the advanced and adjuvant treatment of ACC and sometimes in Cushing’s disease and neuroendocrine ACTH-secreting tumors [3,4]. Currently, it is the only drug approved by the US and European regulatory authorities (FDA, EMA) to treat metastatic ACC. However, it is also used in adjuvant settings [1,3]. This drug represents the medical gold standard in the treatment of ACC, and today, its clinical use has become much more controlled and appropriate. Its mechanism of action, not yet completely understood, inhibits the adrenal cortex’s steroidogenic enzymes, inducing a detrimental effect on the fasciculata and reticularis zonae with a relative effect on the glomerular zone [3]. Although MTT has been used for many years to treat ACC, few data exist on the effects of this drug on other glandular epithelia with steroidogenic activity. In the past, the greatest attention in the care of ACC was directed, above all, to the patient’s overall survival. Currently, the progress made in knowledge about the disease and the improvement of MTT management has considerably prolonged the survival chances of patients. Advances in treatment programs unquestionably require better consideration of those aspects of ACC care considered less relevant in the past. We previously demonstrated that MTT treatment at therapeutic concentrations interferes with follicular development and endocrine ovarian activity without inducing irreversible changes in the gland. Furthermore, a higher drug concentration might lead to increased apoptosis of granulosa cells [5]. Interestingly, data from mating experiments demonstrated that MTT increases breeding time but does not preclude the possibility of procreative function [5]. However, increasing clinical evidence has shown the occurrence of hypogonadism following treatment with this drug, more frequently in male patients [6]. Androgenic deficiency negatively affects bone and muscle metabolism, resulting in many other metabolic complications in men. These complications add to the critical illness resulting from hypercortisolism and oncological disease. On the other hand, the increased survival achieved by young patients should cause us to consider the desire to procreate. Hence, the clinical endocrinologist is called to answer these unanswered questions. To fill this gap, the present study evaluated the impact of MTT treatment on male gonadal function in adult CD1 mice.

## 2. Materials and Methods

### 2.1. Animals

CD1 mice (Charles River, Como, Italy) were housed under controlled temperature (25 °C) and light conditions (12 h light/day) with at libitum access to food and water. At 30 days of age, adult male mice were treated for up to 45 days with 0.152 mg/kg MTT (Sigma-Aldrich Co., St Louis, MO, USA). The length of MTT treatment was chosen to ensure at least one cycle to the seminiferous epithelium (34.5 days in mice), while the drug dose was chosen according to previously published papers [5,7]. The drug was resuspended in dimethyl sulfoxide (DMSO) and administered in a final volume of 20 μL per day/animal via intraperitoneal injection. Healthy mice were treated with the same volume of the vehicle alone (DMSO) (Ctrl). Testosterone replacement therapy, 0.25 mg/mouse/week (Testoviron, Bayer S.p.A., Leverkusen, Germany, 250 mg/mL), was administered by intramuscular injection to a group of MTT-treated adult male animals (MTT + Test) or alone to a control group (Test). The testosterone dose, based on data from Griffin et al. [8], reproduced the physiological levels of testosterone found in male mice [9,10]. The hormone was resuspended in sunflower seed oil (Sigma-Aldrich) in a final volume of 50 µL per week/animal to administer 0.25 mg/mouse/week. The animals were weighed daily and sacrificed by cervical dislocation at the end of treatment. All animal procedures were approved by local ethics committee for animal research.

### 2.2. Quantitative Morphometric Evaluation of Seminiferous Tubules

For morphological analysis, the gonads were quickly removed, fixed in Bouin’s liquid, paraffin-embedded, serially sectioned at 6 µm, and stained with carmalum. Testis sections were examined under a bright-field microscope for quantification of morphological changes. Pictures were taken from every fifth section of the whole testicle. All quantifications were performed on stored images using Image J software. The seminiferous tubule diameter and the area from about 70 sections of seminiferous tubules that were round or nearly round were chosen and measured for each group. The software was calibrated by using the ‘straight line’ icon and by extending a straight line over the length indicating a known distance of 100 µm.

### 2.3. Immunofluorescence

For immunofluorescence analysis, the gonads were quickly removed, fixed in 4% PFA paraffin-embedded, serially sectioned at 6 µm, and mounted on Polysine-TM slides (Menzel-Glaser, Braunschweig, Germany). The slides were incubated with 1 M glycine at a pH of 7.5 for 30 min at room temperature and then in a blocking solution of 1× phosphate-buffered saline (PBS) containing 1% *w*/*v* bovine serum albumin (BSA; Sigma-Aldrich) and 5% *v*/*v* normal donkey serum (Sigma-Aldrich) to minimize non-specific binding. The slides were then incubated for 20 h at 4 °C with 1:300 anti-3β-HSD goat polyclonal IgG (Santa Cruz Biotechnology, Milano, Italy; sc-30820). After extensive washing in PBS, the slides were incubated for 2 h at room temperature with 1:500 donkey anti-goat IgG (H+L) cross-adsorbed secondary antibody from Alexa Fluor™ 488 (Thermo Fisher Scientific, Rome, Italy). Nuclei were then counterstained with Hoechst 33342, and the slides were closed with Vectashield mounting medium (Vector Laboratories, Inc., Newark, CA, USA). Pictures were acquired and examined using a Zeiss Axioscope Imager 2 fluorescence microscope. In the control samples, the primary antibody was substituted with rabbit pre-immune serum.

### 2.4. Isolation of Interstitial Cells from Seminiferous Tubules

The testes were removed from the scrotum, decapsulated, transferred to small Petri dishes in M2 medium containing 0.18% trypsin, and then transferred to a shaking water bath at 32 °C under agitation (90 cycles/min) for 15 min to detach the interstitium. After dissociation, the enzyme was diluted with medium, and the seminiferous tubules were removed by gravity sedimentation. The tubules were washed again to detach the remaining interstitial cells, and the two supernatants were pooled together. The two fractions corresponding to isolated seminiferous tubules and a supernatant enriched in Leydig cells (LCs) were collected for further analysis.

### 2.5. Serum Testosterone Assay

Blood samples were collected under anesthesia via cardiac puncture from all the mice after 45 days of treatment. Clots were removed by centrifugation at 2000× *g* for 10 min, and the resulting supernatant in serum was collected and stored at −20 °C until required for testosterone assay. Total plasma testosterone concentrations were measured by a radioimmunoassay (RIA) kit (Diagnostic Products, Los Angeles, CA, USA) using ’Coat-a-count’ kits from Siemens Healthcare Diagnostics (Los Angeles, CA, USA) with a sensitivity of 4 ng/dL.

### 2.6. RNA Extraction, Reverse Transcription, and Real-Time PCR 

Total RNA was isolated using a silica gel-based membrane spin column (Rneasy Kit, Qiagen S.p.A., Milan, Italy) from the whole gonads. Total RNA was reverse transcribed using the M-MLV reverse transcriptase kit (Qiagen). Ribonucleic acid integrity and purity were confirmed spectroscopically and by gel electrophoresis before use. Total RNA (1 μg) was reverse transcribed in a final volume of 30 μL using the M-MLV Reverse Transcriptase kit (Invitrogen, Milan, Italy); cDNA was diluted 1:2 in nuclease-free water, aliquoted, and stored at −20 °C. The presence of transcripts for insulin-like 3 *(Insl3*), 3β-hydroxysteroid dehydrogenase/isomerase type 1 (*3β-Hsd1*), and follicle stimulating hormone receptor (*Fsh-R)* was evaluated by SYBR Green real-time PCR on an Applied Biosystems 7500 real-time PCR system (Thermo Fisher Scientific, Rome, Italy), equipped with 96-well optical reaction plates using SYBR Green Universal PCR Master Mix (Euroclone, Milan, Italy), by adding 0.3 μmol/L of each specific primer to a total volume of 20 μL reaction mixture according to the manufacturer’s recommendations; negative controls contained water instead of first-strand cDNA. Each sample was normalized to its *β-actin* content. The results were expressed as arbitrary units (a.u.) calculated using the ΔΔCt method. The primers used are shown in Appendix A.

### 2.7. Sperm Extraction and Flow Cytometry Sperm Chromatin Structure Assay

For the MTT-induced DNA damage to sperm, a sperm chromatin structure assay was performed, which evaluates low pH-induced denaturation of sperm nuclear DNA [8]. The right and left epididymis cauda were dissected, transferred individually into small Petri dishes, minced in M2 medium, and incubated for 15 min at 37 °C (95% air and 5% CO_2_) to allow for the release of sperm for the determination of epidydimal sperm reserve (ESR). Samples were diluted in a final volume of 1 mL, and the sperm concentration was evaluated using a hemocytometer. The samples were then frozen at −80 °C and thawed at the analysis time. They were diluted to a concentration of 5 × 10^5^ sperm/mL with TNE buffer solution (0.15 M NaCl, 0.01 M Tris-HCL, 1 mM disodium EDTA, pH 7.4). One hundred microliters of diluted samples were loaded onto the sorter in 12 × 7.5 mL BD FACS tubes and mixed with 200 µL of low-pH detergent solution (0.1% Triton X-100, 0.15 M NaCl, 0.08 N HCL, pH 1.2). After 30 s, the cells were stained with 600 µL of acridine orange (AO) diluted in a buffer composed of 0.1 M citric acid, 0.2 M Na_2_HPO_4_, 1 mM EDTA, and 0.15 M NaCl with a pH of 6 until a final concentration of 6 µg/mL was obtained. Acridine orange is a cell-permeant nucleic acid binding dye that emits green fluorescence when bound to dsDNA and red fluorescence when bound to ssDNA or RNA. Therefore, live cells have a typical green nucleus, and late apoptotic cells display condensed and fragmented orange chromatin [11]. Samples were analyzed using flow cytometry (Dako Cyan ADP, Beckman Coulter Inc., Milan, Italy). Green fluorescence was detected at 515–530 nm, and red fluorescence was measured at 630–650 nm. The result allowed us to obtain a DNA fragmentation index (DFI), and this quantification was performed with FlowJo software (BD Bioscence, Milan, Italy)

### 2.8. Statistical Analyses

All experiments were repeated at least three times, and each experiment was performed at least in duplicate. Statistical analyses were performed using ANOVA followed by the Tukey–Kramer test for comparisons of multiple groups or the two-tailed *t*-test when comparing data derived from two groups. Values with *p* < 0.05 were considered statistically significant.

## 3. Results

### 3.1. Mitotane Interferes with Testosterone Production

We did not observe differences in food intake with the different treatments; however, MTT-treated animals displayed a smaller weight increase per day compared to control mice, although the difference was not statistically significant. The replacement of testosterone reversed this decrease (Figure 1a).

Serum testosterone levels in control mice, 5.8 + 1.4 ng/mL, were in line with normal levels found in mice [9,10]. Serum testosterone was significantly lower in MTT-treated animals compared to their control siblings (*p* < 0.05; Figure 1b), confirming the effect of the pharmacological treatment on steroid production. The testosterone replacement therapy restored the serum hormonal concentrations to levels comparable to those of untreated animals. The group of animals treated with the MTT-testosterone in combination did not differ from that treated with testosterone alone used as an internal control.

### 3.2. MTT Induces Detrimental Effects on Seminiferous Tubules

We next analyzed the impact of MTT treatment on testis morphology. The histological analysis of Ctrl testes showed seminiferous tubules characterized by a well-defined lumen and complete spermatogenesis (Figure 2a,a’ Ctrl). In contrast, testes from the MTT group showed several degrees of damage: disorganization of the germinal epithelium and a pronounced alteration of the spermatogenic process with a reduction in spermatozoa. In some instances, we observed an absence (Figure 2c,c’; arrowhead) or an enlargement (Figure 2b,b’; asterisk) of the lumen, with a less compact interstitial space (Figure 2b–c’). Morphometric analysis revealed that the diameter and area of the seminiferous tubules of treated animals were slightly smaller but not significantly different from those of control animals. Interestingly, in the MTT-Test group, we did not observe gross alterations in tubule morphology (Figure 2d,d’), indicating that the testosterone supplementation counteracted the MTT-induced damage to seminiferous tubules.

### 3.3. Different Effect of MTT on Sertoli and Leydig Cell Gene Expression

To investigate MTT-dependent changes in germ and somatic cells, we evaluated specific gene expression after isolating interstitium and seminiferous tubules as described in the Materials and Methods section. To validate the efficacy of the seminiferous tubule and interstitium separation method, we evaluated by real-time PCR the presence of mRNA for *Fsh-R* and *Insl3*, specific markers of Sertoli (SCs) and LCs, respectively (Appendix A). As shown in Appendix A, *Insl3* mRNA was found mainly in the interstitium. In contrast, *Fsh-R* mRNA was found in the tubules, demonstrating that we efficiently separated the two cell compartments.

Then, we examined the effects of MTT on the expression levels of *3β-Hsd1, Insl3*, and *Fsh-R.* FSH is required for SC proliferation, maturation, and synthesis of specific SC proteins. It is also important at the beginning of the first wave of spermatogenesis [12]. The expression of mRNA from *Fsh-R* was unaffected by treatment, proving that SCs are not the main target of the drug and that the alteration of spermatogenesis is unrelated to SC damage (Figure 3c).

The molecular analysis of the interstitial compartment revealed the expected significant decline in mRNA expression of *Insl3* (*p* < 0.01) and *3β-Hsd1* (*p* < 0.05) in MTT-treated mice, confirming the alteration of the cells responsible for androgen production (Figure 3a,b). The expression of *3β-Hsd1* and *Insl3* in the interstitial cells of testosterone-cotreated mice was partially restored compared to those receiving MTT alone (Figure 3a,b).

Immunofluorescence analysis performed on testicular sections stained with 3β-HSD antibody revealed a less intense fluorescence signal and fewer stained LCs in the MTT-treated animals and recovery after testosterone cotreatment (Figure 3d).

### 3.4. MTT Negatively Affects Sperm Concentration and Quality

The concentration of spermatozoa collected from the epididymis cauda was significantly lower in MTT-treated animals than in the control group (*p* < 0.001; Figure 4a). Moreover, the administration of MTT caused a significant increase in the percentage of spermatozoa with abnormal chromatin structures defined by the DNA fragmentation index (%DFI) compared to the untreated animals (*p* < 0.001; Figure 4b). Significant protection was conferred by testosterone replacement therapy against the MTT-induced sperm damage; in fact, both sperm concentration and viability were restored after testosterone administration (Figure 4a,b).

### 3.5. MTT Withdrawal Restores Testosterone Level but Not the Spermatogenesis

Next, we asked whether the MTT-induced effects were permanent or could be reversed by MTT withdrawal. To this end, after 45 days of treatment, the Ctrl and MTT groups were left for a further 30 days without any treatment and defined as the Ctrl-R and MTT-R groups, respectively. Animal weight was recorded every other day, as well as food intake. After 30 days, one testis/animal was collected for morphological examination and the contralateral testis for RNA extraction. A blood sample from each animal was retrieved to evaluate serum testosterone levels.

We observed that specific genes for the LCs *Insl3* and *3b-Hsd1* were comparable in Ctrl-R and MTT-R mice (Figure 5a,b), as well as testosterone levels (Figure 5c). However, the number of spermatozoa remained statistically lower in MTT-R mice compared to Ctrl-R mice (*p* < 0.01; Figure 5d). In addition, cells with abnormal chromatin structures, calculated by the evaluation of the parameter DNA fragmentation index (%DFI), were significantly more common in MTT-R mice (*p* < 0.01; Figure 5e). Morphological evaluation of seminiferous tubules in the MTT-R mice confirmed these latter data. Although the morphology of the seminiferous tubules was partially recovered, we observed a smaller number of spermatozoa in the tubule sections (Figure 5f–h).

## 4. Discussion

Despite the high aggressiveness of ACC, there have been considerable advances in knowledge about this tumor over the years, and overall survival from it is increasing. This evidence leads us to consider many other parameters during disease management. Unfortunately, information regarding the effects on male testes and fertility is still poor. Many studies have supported evidence of toxic side effects of MTT involving both metabolic and endocrine systems that may require treatment [13,14,15,16]. In a retrospective study in 50 Danish patients, Vikner et al. showed that total cholesterol increased significantly after six months of treatment with MTT, raising LDL, HDL, and triglycerides [17]. Plasma thyroxine decreased while TSH remained unchanged. Drug discontinuation restored the average hormonal balance [17].

We previously demonstrated that MTT at therapeutic concentrations in female mice interferes with follicular development and endocrine ovarian activity. We observed a decrease in early antral follicle numbers with a subsequent increase in secondary follicles. This effect was accompanied by altered steroidogenesis with a significant decrease in mRNA expression of *P450scc* (*Cyp11a1*), which catalyzes the conversion of cholesterol to pregnenolone, and in 17-alpha-hydroxylase (*Cyp17a1*), a key enzyme in the biosynthesis of sex hormones localized in the endoplasmic reticulum of thecal cells. Ovulation was also significantly impaired, with a reduction in the numbers of oocytes, corpora lutea, and pups per litter in the treated animals [5].

In contrast, in males, the available data supporting the harmful effects of MTT have been conflicting. Gentilin et al. demonstrated an inhibitory effect of MTT in mouse gonadotroph cell lines [15]. According to the authors, MTT reduced cell viability, induced apoptosis, and modified cell cycle phases and secretion in gonadotroph cells. These effects could explain the lack of LH increase during MTT treatment despite the reduction in testosterone levels [15]. Moreover, an increase in sex hormone-binding globulin (SHBG) and cortisol-binding globulin (CBG) has been shown in males treated with MTT, ranging from 4 to 6.5 daily for recurrent ACC [18]. It has also been reported that SHBG significantly increases in the first six months after MTT treatment, followed by a progressive decline in free testosterone concentrations and unchanged gonadotropin levels [13,16].

This study aimed to evaluate the effect of MTT on the male gonad, either on androgen hormone synthesis or on testis morphological structure. In addition, possible detrimental involvement in sperm parameters was evaluated.

Animals well tolerated the treatment with MTT, and no changes in weight were observed (Figure 1a). However, MTT treatment significantly reduced the plasma levels of total testosterone (*p* < 0.05; Figure 1b). On the contrary, androgen replacement therapy reverted the testosterone drop in MTT-treated mice by equilibrating the hormone level comparable to mice treated with testosterone alone, used as an internal control. Morphometric analysis revealed that the area and the diameter of seminiferous tubules of the treated animals were slightly smaller than the control, even if not statistically significant. However, the testicular tissue of MTT-treated mice displays different degrees of morpho-structural damage consistent with disorganization of germinal epithelium, either absence or enlargement of the lumen, and a less compact interstitial space (Figure 2). Many data, in the past, recognized in a variety of species, various degrees of testicular damage, followed by a decline in androgen synthesis and an impairment of spermatogenesis by DDT (1,1 bis (p-chlorophenyl) 2,2,2 trichloroethane), an organochlorine insecticide, from which the MTT derives [19,20].

Data relating to the biochemical trend of testosterone during treatment of the patients have been inconsistent, but all agree in attributing clinical overt hypogonadism to many of them. A prospective study observed a biphasic trend of total testosterone levels in seven male patients enrolled to evaluate MTT toxicity after adrenalectomy for ACC [18]. It was characterized by a sharp increase in the first three months of treatment, followed by a decrease in testosterone after six months. Instead, free testosterone levels were significantly reduced [13]. Basile et al. [16], in a report describing unwanted hormonal and metabolic effects of MTT in an adjuvant setting, reported hypogonadism in 12 of 35 male patients (34.3%) who required testosterone replacement therapy. A more recent paper supported these data, showing an increase in total testosterone in the first six months, followed by a decrease over time in 24 patients during a follow-up over eight years in a retrospective, longitudinal study [6].

In this regard, we evaluated the effects of MTT on the different cellular compartments—the interstitial tissue with LCs and the seminiferous epithelium with SCs and germ cells—to better understand the cellular compartments on which MTT acts. As expected, we observed a significant decline in mRNA expression of specific markers of LCs, including *3β-Hsd1*, a key enzyme involved in the biosynthesis of steroid hormones [21], and *Insl3* [22] in MTT-treated mice, confirming a detrimental impact by MTT on LCs, as shown in Figure 3a,b. These results are in accordance with immunofluorescence analysis performed on the testicular sections stained with 3β-HSD antibody, revealing a less intense fluorescence signal and fewer LCs stained in MTT-treated animals (Figure 3d). The expression levels of *3β-Hsd1* and *Insl3* were partially restored by administering testosterone compared to those in mice receiving MTT alone (Figure 3a,b), highlighting the protective role of testosterone. Conversely, we demonstrated that *Fsh-R* in SCs was unaffected after MTT administration, suggesting that these cells were not the main target of the drug and that the alteration of spermatogenesis was unrelated to SC damage (Figure 3c).

The present findings support, for the first time in an animal model, a primary effect of MTT on the testis and a steroidogenic process regarding what occurs in adrenal [23] and ovarian steroidogenesis [5].

Androgens are essential for male fertility and the maintenance of spermatogenesis. In the testis, LCs produce testosterone, and SCs are the primary target for this androgen, which is required to support germ cell development. Decreased levels of testosterone cause impairment of spermatogenesis and prevent the progression of germ cells to meiosis. Cell counts from MTT-treated animals demonstrated few spermatozoa in the tubular lumen. In addition, the sperm concentrations collected from the tail epididymis were significantly lower (*p* < 0.001; Figure 4a). Furthermore, MTT administration caused a significant increase in the percentage of sperm with abnormal chromatin structures, defined by the DNA fragmentation index (%DFI), compared to untreated animals (*p* < 0.001; Figure 4b). Indeed, testosterone replacement therapy allowed for recovery of cell number concentrations, demonstrating significant protection against MTT-induced sperm damage. These data support the well-recognized role of testosterone as an essential factor for spermatogenesis and male fertility [24,25].

Data from the literature and those presented here support the following case report describing a 41-year-old black man treated with MTT for ectopic ACTH-syndrome who developed erectile dysfunction with a biochemical picture of hypergonadotropic hypogonadism (low testosterone and elevated plasmatic gonadotropins at baseline and an exaggerated response to gonadotropin-releasing hormone at dynamic testing) and testes reduced in size. Testicular biopsy showed damage to the seminiferous epithelium with maturation arrest and azoospermia. Administration of testosterone enanthate improved libido and the ability to produce semen [26]. All the data collected so far allow us to hypothesize that MTT acts by inducing a direct toxic effect on the male gonad and through the reduction in androgen hormone synthesis.

Finally, after one-month discontinuation of MTT treatment, we showed recovery of the MTT-induced detrimental effects on the testes. The expression of LC-specific genes, *Insl3 and 3β-Hsd1,* in MTT-R mice was comparable to the levels in Ctrl-R mice. Moreover, complete recovery of testosterone levels and initial recovery in epididymal sperm numbers were observed. This outcome suggests that, as for females, the MTT-induced damage to male gonads could be reversed by MTT withdrawal [5].

The present study and the previous ones that described the indirect effects strongly support the evidence that MTT can induce hypogonadism in males. We can hypothesize that the mechanism through which the drug exerts its action is twofold, with a first acute phase aimed directly at the testis with a harmful effect, either on the germinal and tubular epithelium or on the LCs, with a negative impact on androgen production. Furthermore, the evidence from our study demonstrates that MTT induces an impairment of spermatogenesis with increased DNA fragmentation. These data agree with those observed during clinical practice [26]. Furthermore, they provide strong evidence of the risk to sperm quality in patients treated with MTT, opening the consideration of semen cryopreservation before starting therapy for those patients planning to procreate. A second indirect mechanism, which would explain the hypogonadism induced by MTT, occurs at a later time after prolonged administration of MTT. It predicts a dual effect at the level of the pituitary gonadotropic cells and at the liver, resulting in an inhibitory gonadotropin release [15] and in an increase in the circulating level of SHBG [6,13,16,18]. These conditions may lead to a progressive decline in free testosterone and a paradoxical reduction in LH levels. These data account for the effects of MTT clinically observed during therapy in patients with ACC and hypercortisolism and for measures to be taken before and after MTT therapy. However, to support this recommend, future clinical studies in male patients are needed to confirm the findings.

## 5. Conclusions

For the first time, our study analyzed the effects of MTT treatment on the male reproductive system using an in vivo model. It demonstrated that MTT induces adverse effects on male mouse gonads, including changes in testis morphology and reduced sperm concentration and quality.

## Figures and Tables

**Figure 1 cancers-15-03234-f001:**
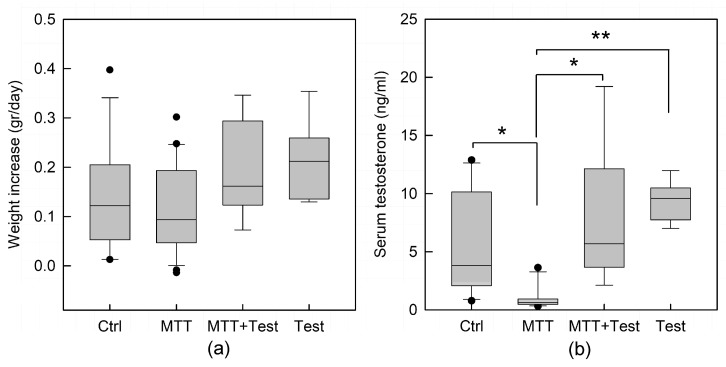
Effect of mitotane on: (**a**) daily body weight increase; and (**b**) serum testosterone dosage in control (Ctrl), mitotane (MTT), MTT-testosterone cotreated (MTT + Test), and testosterone (Test) treated animals. The values are expressed as the mean ± s.e.m. with a total animal number of C = 10; MTT = 11; M + T = 6; Test = 6. Statistical analysis was performed by one-way ANOVA, followed by the Tukey–Kramer test. * *p* < 0.05; ** *p* < 0.01.

**Figure 2 cancers-15-03234-f002:**
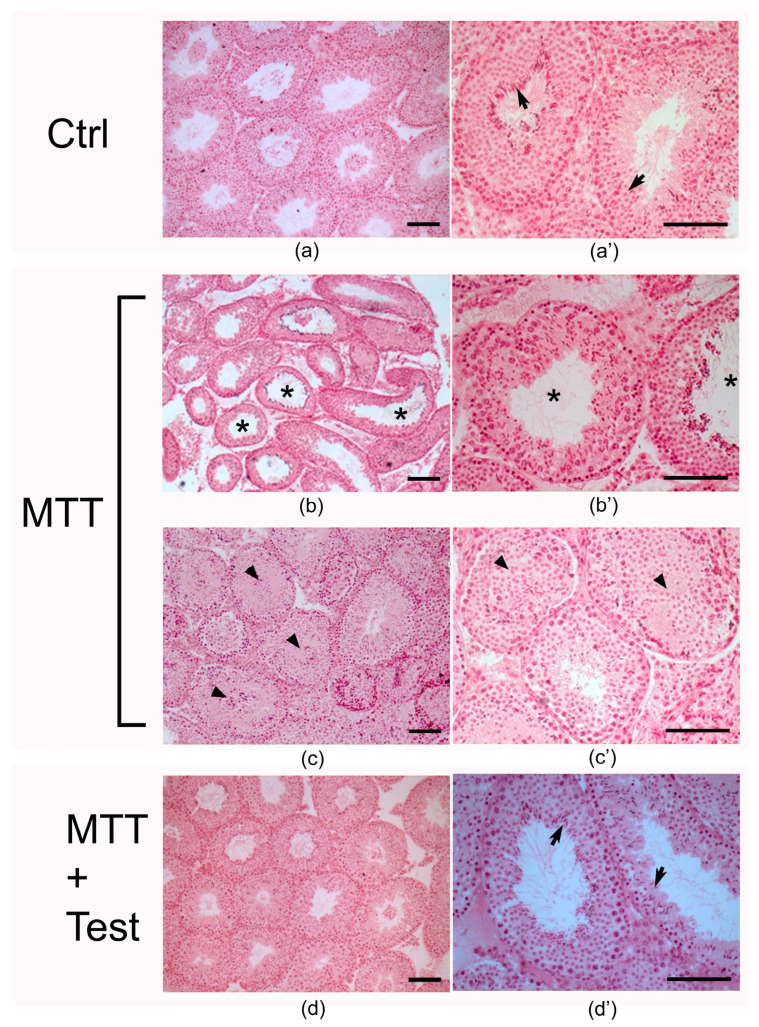
Photomicrographs of sections of testes from (**a**,**a’**, Ctrl) control, (**b**–**c’**, MTT) MTT-treated, and (**d**,**d’**, MTT + Test) MTT and testosterone-treated mice. Arrows: spermatozoa; asterisks: tubules with enlarged lumen; arrowheads: tubules with absence of lumen. Scale bar = 100 μm.

**Figure 3 cancers-15-03234-f003:**
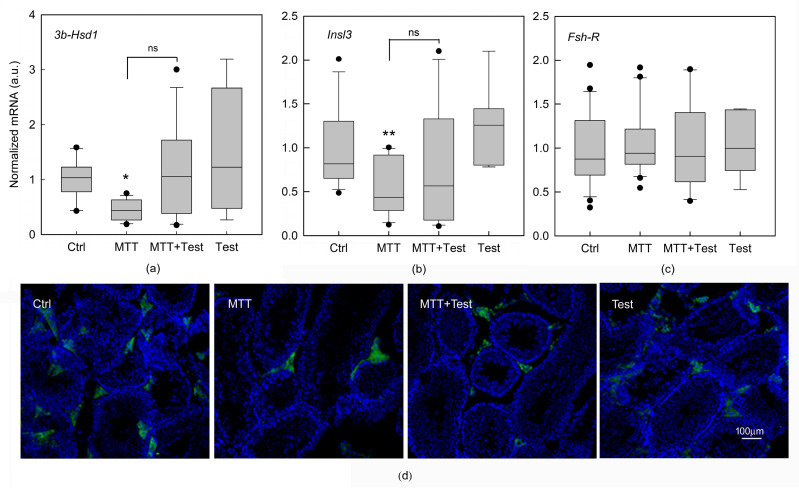
Effects of mitotane on gene expression in isolated seminiferous tubules and Leydig cells removed from untreated mice (Ctrl) and mice treated with mitotane (MTT), MTT and testosterone (MTT+Test) together, and testosterone (Test). Expression of *3β-Hsd1* (**a**) and *Insl3* (**b**) in LCs, detected by real-time PCR in LCs within interstitial cells. Expression of *Fsh-R* (**c**) mRNA in mouse SCs within the seminiferous tubules detected by real-time PCR. Each sample was normalized to its *β-actin* content. Results are expressed as arbitrary units (a.u.) and are represented as the mean ± s.e.m. of three independent experiments with total animal numbers of Ctrl = 15, MTT = 18 and MTT + Test = 12, and testosterone = 7. Statistical analysis was performed using ANOVA followed by the Tukey–Kramer test; * *p* < 0.05 and ** *p* < 0.01, ns: no Significance. Immunofluorescence analysis (**d**). Representative images of immunofluorescent 3β-HSD staining in sections of testes from mice treated as described above. Scale bar = 100 μm.

**Figure 4 cancers-15-03234-f004:**
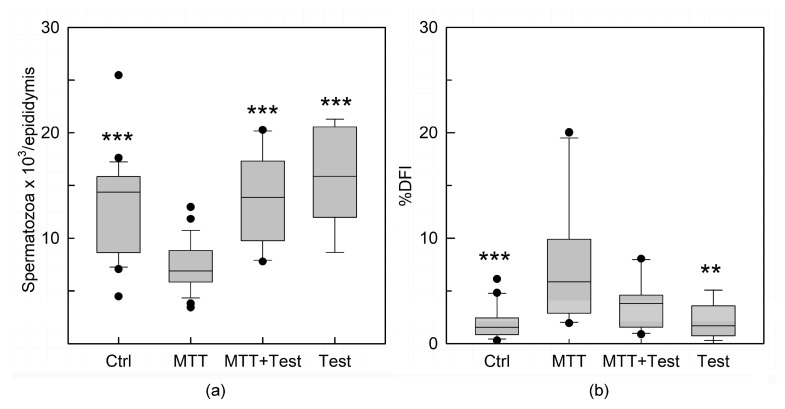
Mean of the number of spermatozoa in mice injected with DMSO (Ctrl), MTT, MTT and testosterone together (MTT + Test) and testosterone (Test). Results are expressed as number of spermatozoa per epididymis (**a**). Quantification of sperm with abnormal chromatin structures calculated by the evaluation of the parameter DNA fragmentation index (DFI). A high %DFI correspond to a large percentage of damaged germ cells. Results are expressed as arbitrary units (a.u.) (**b**). Values are represented as the mean ± s.e.m. of three independent experiments with total animal numbers of Ctrl = 12, MTT = 12, MTT + Test = 7 and Test = 4. Statistical analysis was performed using ANOVA followed by the Tukey–Kramer test. ** *p* < 0.01, *** *p* < 0.001 vs. MTT.

**Figure 5 cancers-15-03234-f005:**
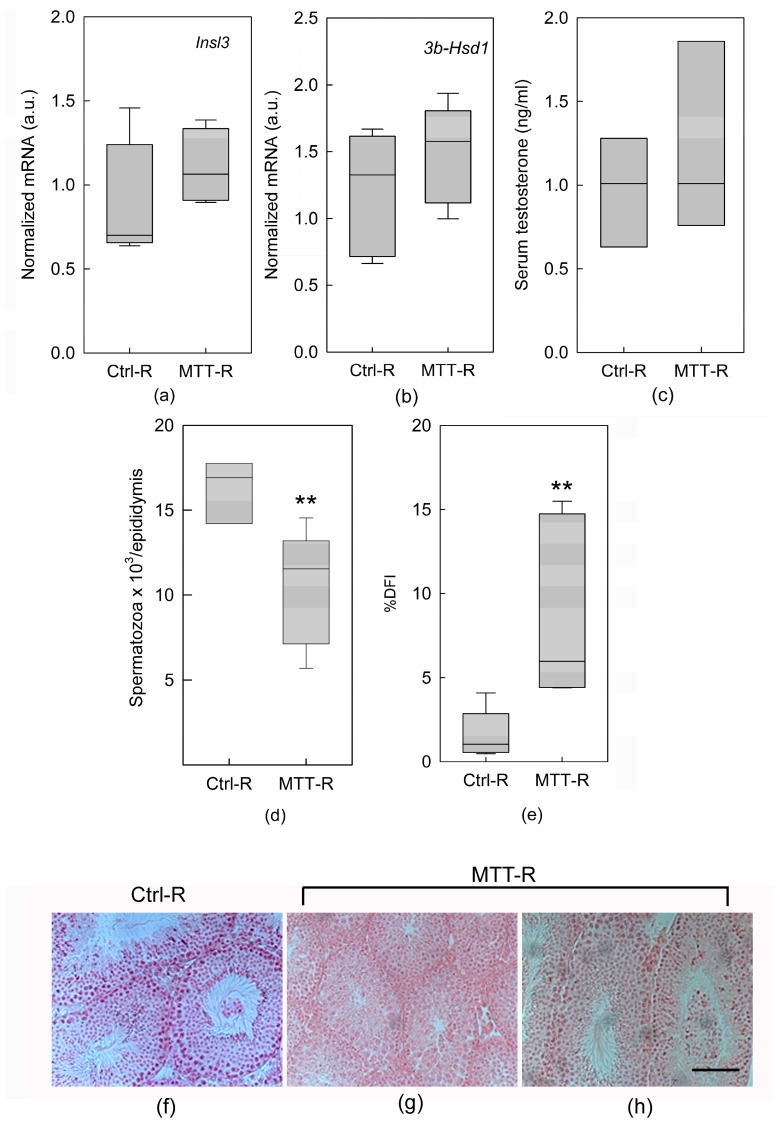
Effect of MTT withdrawal on the expression of *Insl3* (**a**) and *3β-Hsd1* (**b**) in LCs removed from untreated (Ctrl) and mitotane (MTT)-treated mice at 30 days after MTT removal, detected by real-time PCR in LCs within interstitial cells. Each sample was normalized to its *β-actin* content. Results are expressed as arbitrary units (a.u.) and are represented as the mean ± s.e.m. of three independent experiments with a total animal number of Ctrl = 6 and MTT = 6. Serum testosterone concentration (**c**). Mean of the number of spermatozoa (**d**). Quantification of sperm with abnormal chromatin structures calculated by the evaluation of the parameter DNA fragmentation index (DFI) (**e**). A high %DFI corresponds to a large percentage of damaged germ cells. Statistical analysis was performed using Student’s *t* test. ** *p* < 0.01 vs. respective Ctrl-R. Testis morphology after MTT withdrawal (**f**–**h**). Scale bar = 100 mm.

## Data Availability

The data presented in this study are available on request from the corresponding author.

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
