# Peer review of "Effect of Mitotane on Male Gonadal Function"

_cancers, 2023, doi:10.3390/cancers15123234_

Round 1

Reviewer 1 Report

The Ms by Stigliano and coll  entitled:  "Effect of mitotane on male gonadal function" is interesting, as it is known that in ACC male patients, mitotane therapy exposes to elevated risk of hypogonadism that has to be detected, as it has a negative impact on quality of life.

Here Authors characterize the mitotane-induced gonadal dysfunction using with a preclinical in vivo model, namely adult male CD1 mice.

The hypothesis is well defined and the conclusions are supported by results shown.

However, there are some criticisms that need to be resolved.

First of all, Authors should up-grade the references: as examples, but not limited to: doi: 10.1016/j.annonc.2020.08.2099; in particular I would suggest: doi: 10.3389/fendo.2023.1128061: Androgen serum levels in male patients with adrenocortical carcinoma given mitotane therapy: A single center retrospective longitudinal study )

Methods:

Authors need to justify the drug concentration and the length of treatment in mice, both for mitotane and testosterone (please note that the ref 6 and 7 did not reported the schedule applied in the present Ms). Please specify

It is not clear to me the significance of the two animal groups treated, respectively, with the combo mito/testosterone or saline/testosterone. The mito/testosterone group could represent the indirect demonstration of mitotane effect on testicular function and this could be fine, while I do not understand the supplementation with testosterone in normal wild-type male mice. Further, the possible significance of  results obtained in these two groups are not discussed.

The Ms needs a moderate english re-styling.

Author Response

Reviewer 1

  1. First of all, Authors should up-grade the references: ….

The references have been updated; doi:10.1016/j.annonc.2020.08.2099, reference #1; doi: 10.3389/fendo.2023.1128061, reference # 6; according to the referee's suggestions.

  1. Authors need to justify the drug concentration and the length of treatment in mice, both for mitotane and testosterone (please note that the ref 6 and 7 did not reported the schedule applied in the present Ms). Please specify
  • For mitotane’s treatment, drug concentration was chosen according to previously published papers [5,7]. Please note that in the revised version we have added a new reference in the Material and Methods section (Ref #7).
  • The length of mitotane treatment (45 days) was chosen to ensure at least one cycle of the seminiferous epithelium (34.5 days in mice). A sentence was added in the Material and Methods, lines 113-114.
  • The old reference #6 was eliminated and the reference #7 is now #8.
  • As for testosterone (T) supplementation, based on the data from Griffin et al., (Ref #8) we performed a preliminary study to establish which T concentration repristinate the normal T levels in MTT treated animals.
  1. It is not clear to me the significance of the two animal groups treated, respectively, with the combo mito/testosterone or saline/testosterone. The mito/testosterone group could represent the indirect demonstration of mitotane effect on testicular function and this could be fine, while I do not understand the supplementation with testosterone in normal wild-type male mice. Further, the possible significance of results obtained in these two groups are not discussed.

The group of WT mice treated with testosterone (T) represented an internal control to monitor the effect of exogenous T supplementation of the T plasma level. Our results show that T supplementation in WT mice did not alter the T plasma level compared to untreated animals. Please note that in the revised manuscript we modified lines 253-255 in Result section and lines 419-421 in the Discussion section.

  1. The Ms needs a moderate english re-styling.

English quality has been improved.

Reviewer 2 Report

The authors describe the effect of mitotane on gonadal function on different level, a topic which is highly relevant. The methods are sufficient described and the setting is appropriate.

Nevertheless, some minor points should be considered:

-expected normal testosterone level in mice should be mentioned and shown (e.g. horizontal line ) in fig. 1 (b)

-it would have been very informative to measure additional parameters concerning the hormonal axis (LH, FSH). Why were these measurements not performed?

-in addition, the measurement of estradiol and dihydrotestosterone would have been important. Why were these measurements not performed?

-in the discussion part:  the possible mechanism of the "restoring effect" of testosterone replacement should be discussed more in detail.

the quality is sufficient.

Author Response

Dear Editor,

thanks for your e-mail concerning our revised version of manuscript ID 2396992 entitled: “Effect of mitotane on male gonadal function” by Federica Innocenti, Sara Di Persio, Marilena Taggi, Roberta Maggio, Pina Lardo, Vincenzo Toscano, Rita Canipari, Elena Vicini, Antonio Stigliano.  

We have carefully read the criticisms raised by reviewers and their comments. We answer in detail to reviewers 1 and 2 below as you suggested:

Reviewer 2

  1. expected normal testosterone level in mice should be mentioned and shown (e.g. horizontal line) in fig. 1 (b)

We thank the reviewer for the suggestion. The Testosterone (T) levels in the control group are within the normal range as described by Griffin et al (Ref #8). We added a sentence mentioning the T levels obtained in control mice (lines 249-250) in Result section. However, we would like to keep the plot as it is to ensure uniform representation of the data throughout the manuscript. Please note that in the revised manuscript we have added two new reference in line 250 (Ref #9 and #10).

  1. it would have been very informative to measure additional parameters concerning the hormonal axis (LH, FSH). Why were these measurements not performed?

This is a very interesting point raised by referee. We agree with the referee's suggestion but unfortunately the serum obtained from animals is limiting for a multiple hormonal assay.

  1. in addition, the measurement of estradiol and dihydrotestosterone would have been important. Why were these measurements not performed?

The reason is the same described above.

  1. in the discussion part: the possible mechanism of the "restoring effect" of testosterone replacement should be discussed more in detail.

In the revised manuscript we have included two new references describing the central role of testosterone in spermatogenesis and male fertility (Discussion section lines 556-557, Ref #24 and #25).

Round 2

Reviewer 1 Report

The revised Ms satisfies the criticisms raised and it can be accepted for publication 

Reviewer 2 Report

The manuscript could be accepted in present form

The manuscript could be accepted in present form